# Real-World Outcomes and Biomarker Analysis Based on Routine Clinical, Laboratory, and Pathologic Parameters in Metastatic or Unresectable Esophageal Cancer Treated with First-Line Anti-PD-1 Plus Fluoropyrimidine and Platinum

**DOI:** 10.3390/cancers17193149

**Published:** 2025-09-28

**Authors:** Jiyun Jeong, Seyoung Seo, Sung-Bae Kim, Joon Seon Song, Hye Ryun Kim, Byoung Chul Cho, Minkyu Jung, Chang Gon Kim, Moonki Hong, Min Hee Hong, Sook Ryun Park

**Affiliations:** 1Department of Oncology, University of Ulsan College of Medicine, Asan Medical Center, 88, Olympic-ro 43-gil, Songpa-gu, Seoul 05505, Republic of Korea; d251401@amc.seoul.kr (J.J.); syseo@amc.seoul.kr (S.S.); sbkim3@amc.seoul.kr (S.-B.K.); 2Department of Pathology, University of Ulsan College of Medicine, Asan Medical Center, 88, Olympic-ro 43-gil, Songpa-gu, Seoul 05505, Republic of Korea; songjs@amc.seoul.kr; 3Division of Medical Oncology, Department of Internal Medicine, Yonsei University College of Medicine, Yonsei Cancer Center, Severance Hospital, 50-1, Yonsei-ro, Seodaemun-gu, Seoul 03722, Republic of Korea; nobelg@yuhs.ac (H.R.K.); cbc1971@yuhs.ac (B.C.C.); minkjung@yuhs.ac (M.J.); inspector@yuhs.ac (C.G.K.); moonkismile@yuhs.ac (M.H.)

**Keywords:** esophageal squamous cell carcinoma, anti-PD-1 immunotherapy, real-world evidence, prognostic factors, risk stratifications

## Abstract

First-line anti-programmed death-1 (PD-1) plus chemotherapy is the current standard for advanced esophageal squamous cell carcinoma, but real-world data remain limited. In this retrospective analysis, combination therapy demonstrated survival outcomes comparable to pivotal trials despite the inclusion of patients with poor performance status. A point-based prognostic score using Eastern Cooperative Oncology Group (ECOG), C-reactive protein (CRP), and programmed death-ligand 1 (PD-L1) combined positive score (CPS) stratified survival risk, and refined PD-L1 CPS grouping revealed a potential graded association between CPS and treatment outcomes, suggesting value in more nuanced PD-L1 assessment in clinical practice.

## 1. Introduction

Esophageal cancer is one of the most aggressive malignancies, ranking sixth in cancer-related mortality worldwide [1]. Squamous cell carcinoma (SCC) is the predominant histologic subtype in Eastern Europe and Asia, whereas adenocarcinoma is more common in North America and Western Europe [2]. Notably, in early-stage (superficial) disease, SCC is associated with earlier lymphatic spread and a poorer prognosis compared to adenocarcinoma, underscoring their distinct tumor biology and the need for different therapeutic approaches [3]. Despite advances in therapy, the prognosis for patients with metastatic or unresectable esophageal cancer remains dismal, with a median overall survival (OS) of less than one year [4,5]. Standard first-line treatment has traditionally comprised fluoropyrimidine and platinum-based chemotherapy. However, the emergence of immune checkpoint inhibitors (ICIs), particularly anti-programmed death-1 (PD-1) inhibitors, has significantly transformed the therapeutic landscape.

Recent clinical trials have demonstrated that combining anti-PD-1 inhibitors with chemotherapy enhances efficacy and prolongs survival in advanced esophageal cancer [6,7,8,9,10]. This synergy is thought to arise from chemotherapy-induced immunogenic cell death and PD-1 blockade-mediated T-cell reinvigoration [11]. However, most evidence supporting this approach stems from controlled clinical trials with stringent eligibility criteria, which may not accurately reflect outcomes in the broader, more heterogeneous real-world patient population. This limitation is particularly significant in esophageal cancer, where a substantial proportion of patients are elderly and have poor performance status or multiple comorbidities, often excluding them from clinical trials [12,13,14].

Real-world studies are thus essential to assess the effectiveness and safety of anti-PD-1 inhibitor-based combination therapies in a more representative patient population. Furthermore, identifying predictive clinical biomarkers remains an urgent need to better select patients most likely to benefit.

This study aimed to evaluate the real-world outcomes of first-line anti-PD-1 inhibitor combined with fluoropyrimidine and platinum chemotherapy in patients with metastatic or unresectable esophageal squamous cell carcinoma (ESCC). Additionally, we performed biomarker analyses using routinely collected clinical, laboratory, and pathologic parameters to identify predictors of efficacy. By bridging the gap between clinical trial results and routine clinical practice, this research seeks to optimize treatment strategies and personalize therapy for this challenging disease.

## 2. Materials and Methods

### 2.1. Patient Population

This retrospective study included consecutive patients with metastatic or unresectable ESCC who initiated first-line therapy with pembrolizumab or nivolumab in combination with 5-fluorouracil (5-FU) or capecitabine plus cisplatin at two tertiary academic centers—Asan Medical Center and Severance Hospital in Seoul, South Korea—between March 2022 and May 2024.

In accordance with the approval granted by the Ministry of Food and Drug Safety (MFDS) in South Korea, pembrolizumab plus chemotherapy was administered exclusively to patients with a programmed death-ligand 1 (PD-L1) combined positive score (CPS) ≥ 10, while nivolumab plus chemotherapy was administered only to those with a PD-L1 tumor proportion score (TPS) ≥ 1% [15,16]. Additional inclusion criteria were as follows: (1) histologically confirmed ESCC; (2) age ≥ 19 years; and (3) the presence of measurable or evaluable lesions according to the Response Evaluation Criteria in Solid Tumors version 1.1 (RECIST 1.1) [17].

Patients were excluded if they (1) received pembrolizumab or nivolumab in combination with 5-FU or capecitabine plus cisplatin as part of an interventional clinical trial; (2) initiated this combination within 6 months following the last administration of any chemotherapy given as neoadjuvant, adjuvant, or definitive chemoradiotherapy, as such cases would be considered second-line therapy; or (3) had another concurrent malignancy requiring active treatment.

### 2.2. Treatment Regimen and Evaluation of Tumor Response and Adverse Events

Patients were treated with regimens identical to those used in the KEYNOTE-590 and CheckMate 648 trials, according to the approval granted by the MFDS in South Korea [6,7]. Pembrolizumab was administered at 200 mg intravenously (IV) every 3 weeks in combination with 5-FU (800 mg/m^2^ IV on days 1–5) and cisplatin (80 mg/m^2^ IV on day 1) on a 3-week cycle. Nivolumab was administered at 480 mg IV every 4 weeks in combination with 5-FU (800 mg/m^2^ IV on days 1–5) and cisplatin (80 mg/m^2^ IV on day 1) on a 4-week cycle. In some cases, capecitabine was used as an oral alternative to infusional 5-fluorouracil for convenience [5]. Treatment was continued until disease progression, unacceptable toxicity, or patient refusal.

Tumor response was evaluated using computed tomography every 2–3 cycles according to the RECIST v1.1 [17]. Adverse events (AEs) were monitored throughout treatment and follow-up, and were graded according to the National Cancer Institute Common Terminology Criteria for Adverse Events (CTCAE) version 5.0 [18]. Safety outcomes included treatment-related adverse events (TRAEs) and immune-related adverse events (irAEs) [19,20].

### 2.3. Clinical Data Collection and Management

Demographic, clinical, laboratory, and pathologic data were retrospectively collected from the electronic medical records of each participating institution. All datasets were independently curated by investigators at the respective sites, and no cross-institutional data linkage was performed, thereby eliminating the possibility of duplicated cases. Data extraction was performed using predefined case report templates and included age, sex, smoking and alcohol history, Eastern Cooperative Oncology Group (ECOG) performance status, disease status (metastatic or locally advanced unresectable), sites of metastasis (lymph node, liver, lung, bone, peritoneum, and others), recent antibiotics use—defined as the administration of at least one dose within one month before ICI initiation—baseline laboratory findings, tumor PD-L1 immunohistochemistry (IHC), tumor mutation burden (TMB), microsatellite instability (MSI) status, prior treatments, treatment regimens, AEs, and treatment outcomes. TMB was assessed using next-generation sequencing (NGS) using the in-house OncoPanel at Asan Medical Center [21] and the TruSight Oncology 500 (TSO-500) panel (Illumina, San Diego, CA, USA) at Severance Hospital [22]. The MSI status of the patients was determined via NGS, IHC, or polymerase chain reaction (PCR), with IHC staining for MLH1, PMS2, MSH2, and MSH6 performed using the OptiView DAB IHC Detection Kit (Roche Diagnostics, Basel, Switzerland) [23], and PCR-based MSI testing conducted using a five-marker panel (BAT25, BAT26, D5S346, D2S123, and D17S250) on formalin-fixed paraffin-embedded (FFPE) tissue [24,25].

Baseline and on-treatment laboratory blood tests included absolute neutrophil count, absolute lymphocyte count, hemoglobin, platelet count, sodium, albumin, lactate dehydrogenase, and C-reactive protein (CRP). The neutrophil-to-lymphocyte ratio (NLR) was calculated by dividing the absolute neutrophil count by the absolute lymphocyte count, and the platelet-to-lymphocyte ratio (PLR) was determined by dividing the absolute platelet count by the absolute lymphocyte count [26,27,28].

PD-L1 IHC staining was performed on tumor samples from primary or metastatic sites using the Dako Autostainer Link 48 system (Agilent Technologies, Santa Clara, CA, USA), with the PD-L1 IHC 22C3 pharmDx (Agilent Technologies, Santa Clara, CA, USA) and/or 28-8 pharmDx kits (Agilent Technologies, Santa Clara, CA, USA). PD-L1 expression was assessed using TPS, CPS, or both. TPS was defined as the percentage of PD-L1-stained tumor cells among all viable tumor cells. The CPS was calculated as the number of PD-L1-stained tumor cells, lymphocytes, and macrophages divided by the total number of viable tumor cells, multiplied by 100, with a maximum score of 100.

Data cleaning involved cross-checking key variables, including treatment start dates, laboratory values, and response assessment dates, to resolve inconsistencies and address missing data. Data completeness was reviewed prior to analysis, and missing variables were explicitly noted. Quality control measures included independent validation by the site principal investigator. The final dataset was locked for analysis on April 4, 2025. Details regarding database governance, ownership, metadata, and data accessibility are provided in Appendix A.

### 2.4. Statistical Analysis

Descriptive analyses were performed to summarize patient characteristics and key clinical parameters.

The objective response rate (ORR) was defined as the proportion of patients achieving a complete response (CR) or partial response (PR), while the disease control rate (DCR) was defined as the proportion of patients achieving a CR, PR, stable disease (SD), or non-CR/non-progressive disease. Duration of response (DoR) was defined as the time from the first documented CR or PR to progressive disease (PD) or death in responders; patients maintaining CR or PR were censored at their last visit. Progression-free survival (PFS) was calculated from the initiation of first-line treatment to the date of disease progression or death from any cause, whichever occurred first. Patients who were alive without PD were censored on the date of their last assessment. OS was defined as the time from the initiation of first-line treatment to death from any cause; surviving patients were censored at the last follow-up.

Survival outcomes were estimated using the Kaplan–Meier method and compared using the log-rank test. Prognostic factors for PFS and OS were evaluated using the Cox proportional hazards regression model. For continuous variables without established normal ranges, optimal cut-off values were determined using the maximally selected log-rank statistic based on PFS [29]. PD-L1 CPS was categorized into three groups (<10, 10–49, and ≥50) to enable refined stratification of survival risk. Multivariable analysis included variables with a *p*-value < 0.05 in the univariable analysis, and a backward elimination method was used to finalize the model.

Among the PD-L1 expression indicators (CPS and TPS), only one was included in the multivariable Cox regression model to avoid multicollinearity. Of the current companion diagnostics used to guide pembrolizumab plus chemotherapy (CPS 22C3) and nivolumab plus chemotherapy (TPS 28-8), CPS (22C3) was selected because it was available for a greater number of patients in this cohort compared to TPS (28-8). For hematologic parameters, absolute neutrophil count, lymphocyte count, and platelet count were not included separately, as including these variables simultaneously in the Cox model could lead to multicollinearity. Instead, NLR and PLR were used, as these composite indices integrate the relevant measures and are widely recognized as comprehensive indicators of systemic inflammation and prognosis. Similarly, variables for organ-specific metastasis (e.g., lung, liver) were excluded because the presence of metastasis in a specific organ was often associated with a higher number of metastatic organs, leading to potential multicollinearity. Instead, the number of metastatic organs (≥2) was used as a composite variable to represent overall metastatic burden and to improve model simplicity and interpretability.

All statistical tests were two-sided, and a *p*-value < 0.05 was considered statistically significant. Statistical analyses were conducted using R software (version 4.4.2; R Foundation for Statistical Computing, Vienna, Austria).

## 3. Results

### 3.1. Patient Characteristics

A total of 131 esophageal cancer patients who received ICI-based chemotherapy at Asan Medical Center (*n* = 78) or Severance Hospital (*n* = 52) between 2022 and 2024 were screened. After applying the eligibility criteria and data cleaning, 87 patients were included in the final analysis (Figure 1). The median age was 62 years (range, 42–87), and 16.1% of patients had an ECOG performance status of ≥2. Most patients (81.6%) had metastatic disease at baseline, and 43.7% had distant metastases involving two or more organs. The most frequent metastatic sites were distant lymph nodes (58.6%), lungs (33.3%), and liver (26.4%) (Table 1).

PD-L1 expression was assessed using both CPS and TPS, with either the 22C3 or 28-8 antibody clones. Some patients had results available for both scoring systems and both antibody clones. CPS (22C3) results were available in 83 patients (95.4%), of whom 77.1% had a CPS ≥ 10. CPS (28-8) results were available in 34 patients (39.1%), of whom 55.9% showed a CPS ≥ 10. TPS (22C3) results were available in 86 patients (98.9%), of whom 79.1% had a TPS ≥ 1%. TPS (28-8) results were available in 50 patients (57.5%), of whom 88.0% showed a TPS ≥ 1%.

The median TMB was 17 mutations/Mb (range, 6–69). MSI status was assessed in 38 patients (43.7%), and 2 patients (2.3%) were classified as MSI-high or mismatch repair-deficient (dMMR) based on NGS, IHC, or PCR.

### 3.2. Treatment Administered

Among the 87 patients included in the study, 60.9% (*n* = 53) received pembrolizumab plus 5-fluorouracil and cisplatin (FP), and 9.2% (*n* = 8) received pembrolizumab plus capecitabine and cisplatin (XP). Three patients received both FP and XP regimens during the course of treatment. Nivolumab was administered to 29 patients, with 31.0% (*n* = 27) receiving nivolumab plus FP and 2.3% (*n* = 2) receiving nivolumab plus XP.

The median duration of treatment was 5.1 months (range, 0.7–29.6) in the pembrolizumab plus chemotherapy group and 4.7 months (range, 0.9–18.2) in the nivolumab plus chemotherapy group (Appendix A). Among patients treated with pembrolizumab, no dose reductions occurred; however, dose delays were observed in 50.0% of patients. A dose delay was defined as initiating the next treatment cycle more than three days later than the scheduled date. Chemotherapy components were frequently modified: cisplatin and 5-FU or capecitabine doses were reduced in 79.3% and 70.7% of patients, respectively, and dose delays occurred in 46.6% and 48.3%. In the nivolumab group, dose reductions occurred in 6.9% of patients, and 41.4% experienced dose delays. For chemotherapy components, cisplatin and 5-FU or capecitabine were reduced in 62.1% and 65.5% of patients, respectively, with dose delays occurring in 44.8% each.

In total, 71 patients (81.6%) discontinued treatment. The primary reasons for discontinuation were disease progression (66.2%), toxicity (9.9%), and patient refusal (7.0%). Details of subsequent therapies are provided in Appendix A.

### 3.3. Safety and Tolerance

TRAEs of any grade occurred in 82 patients (94.3%), with grade 3–4 events reported in 45 patients (51.7%) (Table 2). Serious TRAEs were observed in 4 patients (4.6%), and no grade 5 TRAEs were reported. The most common TRAEs were nausea (37.9%), anemia (27.6%), fatigue (25.3%), alopecia (25.3%), neutropenia (24.1%), peripheral neuropathy (23.0%), and anorexia (20.7%). The most frequent grade 3–4 TRAEs were neutropenia (16.1%), anemia (12.6%), and nausea (8.0%).

irAEs occurred in 26 patients (29.9%). The most frequently reported irAEs were pruritus (13.8%), skin rash (12.6%), dry skin (3.4%), and pneumonitis (3.4%). Most irAEs were grade 1–2; however, one case each of grade 3 immune-related pneumonitis and immune-related cholangiopathy was observed.

### 3.4. Efficacy Outcomes

Appendix A summarizes the treatment efficacy outcomes. The median follow-up duration was 17.8 months (range, 0.3–37.3). Overall, 44 patients (50.6%) died, and 62 (71.3%) experienced disease progression or death during the study period.

Among all 87 patients, the ORR was 48.3% (42 of 87; 95% confidence interval [CI], 37.4–59.2), including 4 patients (4.6%) who achieved a CR and 38 (43.7%) who achieved a PR. The DCR was 77.0% (67 of 87; 95% CI, 66.8–85.3), including patients with CR, PR, SD, or non-CR/non–PD. The median DoR was 11.8 months (95% CI, 6.9–not reached).

The median PFS was 5.6 months (95% CI, 4.5–8.7), with 6-month and 1-year PFS rates of 46.7% (95% CI, 36.9–58.9) and 29.0% (95% CI, 20.4–41.2), respectively (Figure 2A). The median OS was 13.1 months (95% CI, 10.6–not reached), with 6-month and 1-year OS rates of 78.7% (95% CI, 70.5–88.0) and 52.1% (95% CI, 42.0–64.6), respectively (Figure 2B).

### 3.5. Analysis of Clinicolaboratory Factors for PFS and OS

Table 3 summarizes the results of univariable and multivariable Cox regression analyses for PFS. In the univariable analysis, the following factors were significantly associated with shorter PFS: ECOG performance status ≥ 2 (*p* < 0.001), elevated baseline CRP above the upper normal limit (UNL; *p* < 0.001), elevated baseline absolute neutrophil count (≥4000/μL; *p* = 0.002), high baseline NLR (≥3.37; *p* < 0.001), high baseline PLR (≥146.4; *p* < 0.001), liver metastasis (*p* = 0.015), lung metastasis (*p* = 0.028), low baseline sodium (<135 mmol/L; *p* = 0.002), recent antibiotic use (*p* = 0.036), and lower PD-L1 expression based on CPS 22C3 (<10 or 10–49 vs. ≥50, both *p* = 0.002). In the multivariable analysis, three factors remained independently associated with shorter PFS: ECOG performance status ≥ 2 (hazard ratio [HR], 4.55; 95% CI, 2.00–10.34; *p* < 0.001), elevated baseline CRP above UNL (HR, 2.44; 95% CI, 1.20–4.96; *p* = 0.013), and lower PD-L1 expression based on CPS 22C3 <50 (10–49 vs. ≥50: HR, 2.46; 95% CI, 1.01–6.00; *p* = 0.047; <10 vs. ≥50: HR, 4.14; 95% CI, 1.51–11.32; *p* = 0.006).

Table 4 presents the results for OS. In the univariable analysis, the following factors were significantly associated with shorter OS: ECOG performance status ≥ 2 (*p* < 0.001), liver metastasis (*p* < 0.001), bone metastasis (*p* = 0.006), multiple metastatic organs (≥2; *p* = 0.024), elevated baseline CRP above UNL (*p* = 0.001), high baseline absolute neutrophil count (≥4000/μL; *p* = 0.018), low baseline albumin (<3.5 g/dL; *p* = 0.010), low baseline sodium (<135 mmol/L; *p* = 0.004), high baseline NLR (≥3.37; *p* = 0.003), high baseline PLR (≥146.4; *p* = 0.004), recent antibiotic use (*p* = 0.009), and low PD-L1 expression based on CPS 22C3 (<10 or 10–49 vs. ≥50; *p* = 0.001 and *p* = 0.013, respectively). In the multivariable analysis, the following factors remained independent predictors of worse OS: ECOG performance status ≥ 2 (HR, 2.90; 95% CI, 1.23–6.85; *p* = 0.015), elevated CRP above UNL (HR, 2.22; 95% CI, 1.06–4.66; *p* = 0.034), and lower PD-L1 expression based on CPS 22C3 < 10 vs. ≥50 (HR, 5.82; 95% CI, 1.78–19.03; *p* = 0.004). CPS 10–49 showed a trend toward shorter survival compared to CPS ≥50 (HR, 2.28; 95% CI, 0.74–7.00; *p* = 0.152), although this difference was not statistically significant, possibly due to the limited sample size.

### 3.6. Development of a Prognostic Scoring Model and Risk Stratification

A prognostic scoring model was developed based on independent predictors of PFS identified in the multivariable Cox regression analysis: ECOG performance status ≥ 2, elevated baseline CRP, and PD-L1 CPS 22C3 status. For CPS 22C3, values of 10–49 and <10 were each associated with worse outcomes compared to CPS ≥ 50 and were thus assigned separate scores. Integer points were allocated proportionally to the strength of each variable’s regression coefficient. ECOG performance status ≥2 (coefficient = 1.51) and CPS <10 (coefficient = 1.42) were each assigned 2 points, while elevated CRP above UNL (coefficient = 0.89) and CPS 10–49 (coefficient = 0.90) were each assigned 1 point. This simplified scoring system yielded a total score ranging from 0 to 5 and effectively stratified patients by PFS (Appendix A). Patients were categorized into four risk groups based on visual inspection of Kaplan–Meier curves and clinical interpretability: low risk (score 0), intermediate–low risk (score 1), intermediate–high risk (score 2), and high risk (score 3–5). This four-tiered classification demonstrated clear separation in PFS among the groups (Figure 3A). The median PFS was not reached in the low-risk group; 10.5 months (95% CI, 4.9–not reached) in the intermediate–low-risk group; 4.0 months (95% CI, 2.4–not reached) in the intermediate–high-risk group; and 2.1 months (95% CI, 1.2–4.1) in the high-risk group (log-rank *p* < 0.0001).

Although the model was developed based on PFS, the OS curves for five individsal scores (0–5) also showed clear separation (Appendix A), supporting the prognostic relevance of the scoring system. Furthermore, the four risk groups derived from this model demonstrated statistically significant differences in OS, reinforcing its prognostic utility beyond PFS (log-rank *p* < 0.0001) (Figure 3B). The median OS was not reached in either the low-risk or intermediate–low-risk group, whereas it was 10.6 months (95% CI, 5.5–not reached) in the intermediate–high-risk group and 6.6 months (95% CI, 2.9–10.6) in the high-risk group.

## 4. Discussion

This multi-institutional retrospective study evaluated the real-world effectiveness and prognostic factors of first-line anti-PD-1 immunotherapy combined with chemotherapy in patients with metastatic or unresectable ESCC. Our findings demonstrate robust clinical activity, acceptable safety, and key baseline factors associated with survival outcomes, which facilitated the development of a simplified prognostic scoring system.

The median PFS of 5.6 months and median OS of 13.1 months in our cohort are comparable to those reported in pivotal clinical trials. In KEYNOTE-590, among patients with ESCC and PD-L1 CPS ≥ 10, the median PFS and OS were 7.3 and 13.9 months, respectively; for all ESCC patients, these values were 6.3 and 12.6 months, respectively [6]. In CheckMate 648, the median PFS and OS were 6.9 and 15.4 months, respectively, for patients with ESCC and PD-L1 TPS ≥ 1% and 5.8 and 13.2 months, respectively, in all ESCC patients [7]. These benchmarks support the real-world effectiveness of pembrolizumab or nivolumab plus chemotherapy in routine practice. The ORR (48.3%), DCR (77.0%), and median DoR (11.8 months) in our cohort also align with outcomes from KEYNOTE-590 (ORR = 45.0%, DCR = 79.4%, median DoR = 8.3 months) and CheckMate 648 (ORR = 47.4%, DCR = 79.4%, median DoR = 8.2 months).

Notably, 16.1% of patients in our cohort had ECOG performance status ≥ 2—individuals typically excluded from clinical trials. In our study, only 23.0% of patients had ECOG performance status 0, compared to 40% in KEYNOTE-590 and 47% in CheckMate 648. These differences highlight the broader applicability of ICI plus chemotherapy in real-world settings.

Safety outcomes were consistent with prior trials. Grade ≥ 3 TRAEs occurred in 51.7% of patients, most commonly neutropenia, anemia, and nausea. These rates were comparable to those in KEYNOTE-590 (72%) and CheckMate 648 (47%). One-third experienced irAEs, mostly grade 1–2, although grade 3 pneumonitis and cholangiopathy were observed. These results reaffirm the manageable safety profile of ICIs plus chemotherapy.

Multivariable Cox regression identified ECOG performance status ≥ 2, elevated CRP (>UNL), and low PD-L1 expression (CPS 22C3 < 10 or 10–49) as independent predictors of shorter PFS and OS. Notably, CPS < 10 consistently correlated with worse outcomes, reinforcing its prognostic value. Although CPS 10–49 did not reach statistical significance for OS, it tended toward inferior survival compared to CPS ≥ 50, suggesting a graded association between CPS level and treatment outcomes. Our four-tiered prognostic scoring model based on these independent risk factors stratified patients into low-, intermediate–low-, intermediate–high-, and high-risk groups, showing significant differences in PFS and OS. High-risk patients (score 3–5) had a median PFS of only 2.1 months, while the low-risk group (score 0) had not reached median PFS, with a 1-year PFS rate > 80%. Similarly, median OS in the high-risk group was markedly inferior (6.6 months) compared to the low-risk group, for whom the median OS was not reached during follow-up. These differences underscore the prognostic utility of our model, which provides a simple and practical framework to identify patients with dismal outcomes who may require alternative or intensified treatment strategies, and to reassure those with favorable profiles. This tool may assist clinicians in making risk-adapted treatment decisions and guiding patient counseling.

While previous real-world studies proposed prognostic factors in esophageal cancer patients treated with first-line anti-PD-1 inhibitor plus chemotherapy, few adequately assessed the prognostic value of PD-L1 expression—one of the most established biomarkers for predicting ICI efficacy—due to limited data availability [30,31]. In KEYNOTE-590, survival benefit from pembrolizumab plus chemotherapy was evident only in patients with CPS ≥ 10 (median OS 13.5 vs. 9.4 months, HR, 0.62, *p* < 0.0001; median PFS 7.5 vs. 5.5 months, HR, 0.51, *p* < 0.0001), with no benefit in CPS < 10 (median OS 10.5 vs. 10.6 months; median PFS 6.2 vs. 6.0 months) [6]. In CheckMate 648, the benefit from nivolumab plus chemotherapy was likewise confined to higher PD-L1 strata (CPS ≥ 1, ≥5, or ≥10) [7]. Importantly, these studies used binary PD-L1 cutoffs (e.g., CPS < 10 vs. ≥10), which grouped a wide range of PD-L1 expression levels—spanning from intermediate to very high—into a single category, thereby obscuring potential gradations in treatment benefit. For instance, patients with CPS values of 11, 40, or 90 would all be lumped into the same “high” expression group, making it difficult to assess whether increasing PD-L1 levels correlate with increasing benefit from ICI-based therapy. In contrast, our classification (<10, 10–49, and ≥50) revealed a clear graded association between PD-L1 CPS and treatment benefit, with CPS ≥ 50 showing the best outcomes, CPS 10–49 intermediate benefit, and CPS < 10 poorest outcomes. This approach may offer a more nuanced framework for assessing PD-L1 as a biomarker, enhancing risk stratification and treatment personalization.

Several limitations of this study should be acknowledged. The retrospective design introduces inherent selection bias. While PD-L1 expression data using the 22C3 antibody—both CPS and TPS—were available for nearly all patients, data using the 28-8 clones were available only in a subset of patients, as its use became more common following the clinical introduction of nivolumab plus chemotherapy later in the study period. The modest sample size also limits the statistical power of some subgroup analyses, especially for OS comparisons among PD-L1 strata. Additionally, missing data for molecular biomarkers, such as MSI status, restricted their incorporation into the multivariable analysis.

Despite these limitations, our findings provide meaningful insight into the real-world performance of anti-PD-1-based regimens and highlight the potential of simplified prognostic tools in routine clinical practice.

## 5. Conclusions

This study demonstrates the real-world effectiveness of first-line anti-PD-1 therapy plus chemotherapy in metastatic or unresectable ESCC and introduces a clinically useful prognostic scoring model based on routinely available clinical parameters. Prospective validation in larger, independent cohorts is warranted to confirm its predictive value and utility in treatment decision-making.

## Figures and Tables

**Figure 1 cancers-17-03149-f001:**
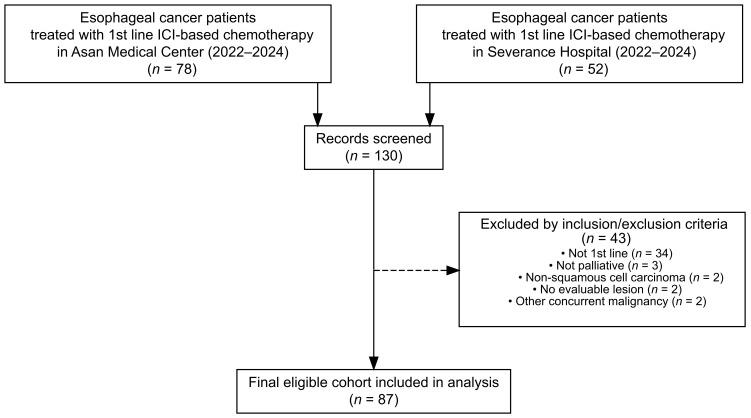
Flow diagram of patient selection and exclusion.

**Figure 2 cancers-17-03149-f002:**
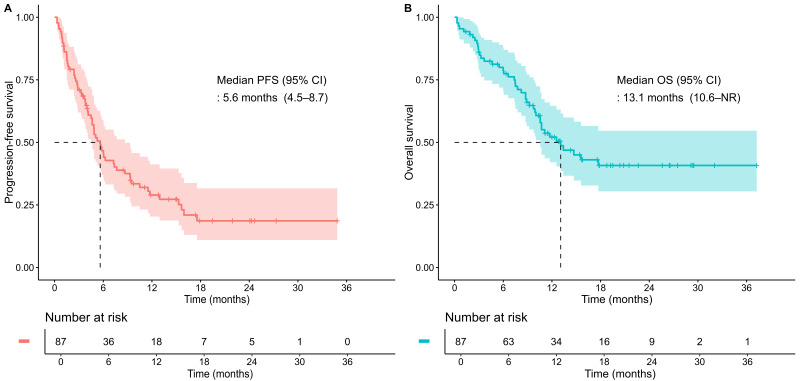
Kaplan–Meier curves for the total study population. (**A**) progression-free survival (PFS); (**B**) overall survival (OS). Tick marks indicate censored observations. Dashed lines denote the median survival time where the survival probability reaches 0.5. CI, confidence interval; NR, not reached.

**Figure 3 cancers-17-03149-f003:**
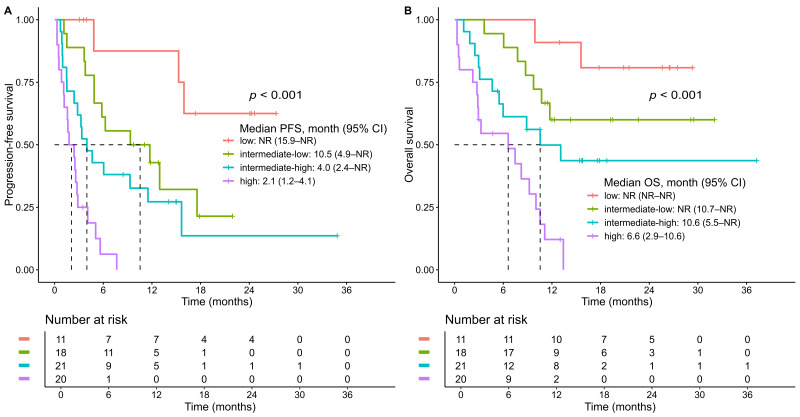
Kaplan–Meier curves for the total study population stratified by risk groups. (**A**) progression-free survival (PFS); (**B**) overall survival (OS). Risk groups were defined as follows: low-risk (score 0), intermediate–low-risk (score 1), intermediate–high-risk (score 2), and high-risk (score 3–5). *p* values were calculated using the log-rank test. Tick marks indicate censored observations. Dashed lines denote the median survival time where the survival probability reaches .0.5. CI, confidence interval; NR, not reached.

**Table 1 cancers-17-03149-t001:** Baseline characteristics.

Characteristics	No. (%) (N = 87)
Age, median (range), years	62 (42–87)
<65 years	55 (63.2%)
≥65 years	32 (36.8%)
Sex	
Male	75 (86.2%)
Female	12 (13.8%)
ECOG Performance status	
0	20 (23.0%)
1	53 (60.9%)
≥2	14 (16.1%)
Smoking Status	
Never smoked	17 (19.5%)
Former smoker	34 (39.1%)
Current smoker	36 (41.4%)
Histological grade	
Well differentiated	7 (8.0%)
Moderately differentiated	55 (63.2%)
Poorly differentiated	15 (17.2%)
Unknown	10 (11.5%)
PD-L1 status	
CPS (22C3) < 10	19 (21.8%)
CPS (22C3) ≥ 10	64 (73.6%)
CPS (22C3) NA	4 (4.6%)
CPS (28-8) < 10	15 (17.2%)
CPS (28-8) ≥ 10	19 (21.8%)
CPS (28-8) NA	53 (60.9%)
TPS (22C3) < 1%	18 (20.7%)
TPS (22C3) ≥ 1%	68 (78.2%)
TPS (22C3) NA	1 (1.1%)
TPS (28-8) < 1%	6 (6.9%)
TPS (28-8) ≥ 1%	44 (50.6%)
TPS (28-8) NA	37 (42.5%)
Disease status	
Metastatic	71 (81.6%)
Locally advanced, unresectable	16 (18.4%)
No. of distant metastatic organs	
0	16 (18.4%)
1	33 (37.9%)
2	23 (26.4%)
≥3	15 (17.2%)
Site of distant metastasis	
Distant lymph node	51 (58.6%)
Lung	29 (33.3%)
Liver	23 (26.4%)
Bone	16 (18.4%)
Peritoneum	4 (4.6%)
Other organs *	5 (5.7%)
Prior treatment for esophageal cancer ^‡^	25 (28.7%)
Surgery	21 (24.1%)
Concurrent chemoradiotherapy	15 (17.2%)
Radiation therapy	2 (2.3%)
Chemotherapy	13 (14.9%)
Recent use of antibiotics ^†^	34 (40.2%)
Tumor mutation burden, median (range), mutations/Mb	17 (6–69)
MSI status ^§^	
MSS/pMMR	36 (41.4%)
MSI-H/dMMR	2 (2.3%)
NA	49 (56.3%)

* Includes metastases to spleen, adrenal gland, kidney, maxilla and surrounding gingiva, thyroid, muscle, pleura, and skin. † Defined as at least one dose of antibiotics administered within 30 days prior to treatment initiation. ‡ The numbers of patients by treatment modality are not mutually exclusive because some patients received more than one prior treatment. § MSI status was determined using next-generation sequencing (NGS), immunohistochemistry (IHC), or polymerase chain reaction (PCR); some patients were tested by more than one method. No discordance was observed among available results. ECOG, Eastern Cooperative Oncology Group; PD-L1, programmed death-ligand 1; CPS, combined positive score; TPS, tumor proportion score; Mb, megabase; MSI, microsatellite instability; MSS, microsatellite stable; pMMR, mismatch repair-proficient; MSI-H, microsatellite instability-high; dMMR, mismatch repair-deficient; NA, not available.

**Table 2 cancers-17-03149-t002:** Treatment-related adverse events.

	CTCAE Grades, No. (%) (N = 87)				
	Any Grade	Grade 1	Grade 2	Grade 3	Grade 4
Any treatment-related adverse events	82 (94.3)	59 (67.8)	68 (78.2)	45 (51.7)	3 (3.4)
Treatment-related serious adverse events	4 (4.6)	0 (0.0)	2 (2.3)	3 (3.4)	0 (0.0)
Treatment-related adverse events (≥5% of patients)					
Nausea	33 (37.9)	10 (11.5)	16 (18.4)	7 (8.0)	0 (0.0)
Anemia	24 (27.6)	0 (0.0)	12 (13.8)	11 (12.6)	1 (1.1)
Fatigue	22 (25.3)	11 (12.6)	9 (10.3)	2 (2.3)	0 (0.0)
Alopecia	22 (25.3)	15 (17.2)	7 (8.0)	0 (0.0)	0 (0.0)
Neutrophil count decreased	21 (24.1)	0 (0.0)	6 (6.9)	14 (16.1)	1 (1.1)
Peripheral neuropathy	20 (23.0)	12 (13.8)	6 (6.9)	2 (2.3)	0 (0.0)
Anorexia	18 (20.7)	3 (3.4)	13 (14.9)	2 (2.3)	0 (0.0)
Platelet count decreased	13 (14.9)	6 (6.9)	5 (5.7)	2 (2.3)	0 (0.0)
Mucositis oral	12 (13.8)	4 (4.6)	6 (6.9)	2 (2.3)	0 (0.0)
Diarrhea	12 (13.8)	6 (6.9)	5 (5.7)	1 (1.1)	0 (0.0)
Skin rash	10 (11.5)	6 (6.9)	4 (4.6)	0 (0.0)	0 (0.0)
Fever	9 (10.3)	7 (8.0)	2 (2.3)	0 (0.0)	0 (0.0)
Vomiting	9 (10.3)	2 (2.3)	5 (5.7)	2 (2.3)	0 (0.0)
Constipation	8 (9.2)	5 (5.7)	3 (3.4)	0 (0.0)	0 (0.0)
Aspartate aminotransferase increased	7 (8.0)	2 (2.3)	3 (3.4)	2 (2.3)	0 (0.0)
Acute kidney injury	7 (8.0)	2 (2.3)	3 (3.4)	2 (2.3)	0 (0.0)
Hand-foot syndrome	6 (6.9)	5 (5.7)	1 (1.1)	0 (0.0)	0 (0.0)
Alanine aminotransferase increased	5 (5.7)	1 (1.1)	3 (3.4)	1 (1.1)	0 (0.0)
Pruritus	5 (5.7)	3 (3.4)	2 (2.3)	0 (0.0)	0 (0.0)
Creatinine increased	5 (5.7)	3 (3.4)	2 (2.3)	0 (0.0)	0 (0.0)
Immune-related adverse events					
Pruritus	12 (13.8)	3 (3.4)	9 (10.3)	0 (0.0)	0 (0.0)
Skin rash	11 (12.6)	4 (4.6)	7 (8.0)	0 (0.0)	0 (0.0)
Dry skin	3 (3.4)	2 (2.3)	1 (1.1)	0 (0.0)	0 (0.0)
Pneumonitis	3 (3.4)	0 (0.0)	2 (2.3)	1 (1.1)	0 (0.0)
Hypothyroidism	2 (2.3)	0 (0.0)	2 (2.3)	0 (0.0)	0 (0.0)
Colitis	2 (2.3)	1 (1.1)	1 (1.1)	0 (0.0)	0 (0.0)
Dry mouth	2 (2.3)	2 (2.3)	0 (0.0)	0 (0.0)	0 (0.0)
Serum amylase increased	2 (2.3)	2 (2.3)	0 (0.0)	0 (0.0)	0 (0.0)
Subclinical hypothyroidism	2 (2.3)	1 (1.1)	1 (1.1)	0 (0.0)	0 (0.0)
Arthralgia	2 (2.3)	1 (1.1)	1 (1.1)	0 (0.0)	0 (0.0)
Arthritis	1 (1.1)	1 (1.1)	0 (0.0)	0 (0.0)	0 (0.0)
Myalgia	1 (1.1)	1 (1.1)	0 (0.0)	0 (0.0)	0 (0.0)
Pericarditis	1 (1.1)	1 (1.1)	0 (0.0)	0 (0.0)	0 (0.0)
Immune-related cholangiopathy	1 (1.1)	0 (0.0)	0 (0.0)	1 (1.1)	0 (0.0)
Thyroiditis	1 (1.1)	0 (0.0)	1 (1.1)	0 (0.0)	0 (0.0)
Subclinical hyperthyroidism	1 (1.1)	1 (1.1)	0 (0.0)	0 (0.0)	0 (0.0)
Nail dystrophy	1 (1.1)	1 (1.1)	0 (0.0)	0 (0.0)	0 (0.0)

Adverse events were graded according to the National Cancer Institute Common Terminology Criteria for Adverse Events (CTCAE), version 5.0.

**Table 3 cancers-17-03149-t003:** The results of univariable and multivariable analyses for progression-free survival.

Variable	Univariable		Multivariable	
	HR (95% CI)	*p* Value	HR (95% CI)	*p* Value
Age (≥65 years vs. <65 years)	0.61 (0.35–1.05)	0.073		
ICI (nivolumab vs. pembrolizumab)	1.29 (0.77–2.19)	0.336		
Sex (female vs. male)	0.86 (0.41–1.82)	0.693		
No. of metastatic organs (≥2 vs. ≤1)	1.91 (1.14–3.21)	0.014 *		
Liver metastasis (yes vs. no)	1.98 (1.14–3.42)	0.015 *		
Lung metastasis (yes vs. no)	1.79 (1.06–3.02)	0.028 *		
Peritoneum metastasis (yes vs. no)	1.71 (0.62–4.74)	0.301		
Bone metastasis (yes vs. no)	1.81 (0.96–3.42)	0.066		
Lymph node metastasis (yes vs. no)	0.86 (0.52–1.42)	0.550		
Cigarette use (never vs. former/current smoker)	0.84 (0.46–1.55)	0.584		
ECOG PS (≥2 vs. 0–1)	5.20 (2.67–10.12)	<0.001 *	4.55 (2–10.34)	<0.001 *
Tumor mutation burden (≥20 Mb vs. <20 Mb)	0.87 (0.38–1.98)	0.737		
Baseline Hb (≥12 g/dL vs. <12 g/dL)	0.69 (0.42–1.14)	0.151		
Baseline ALC (≥1000/μL vs. <1000/μL)	0.55 (0.28–1.09)	0.086		
Baseline ANC (≥4000/μL vs. <4000/μL)	2.69 (1.45–4.97)	0.002 *		
Baseline CRP (> UNL vs. ≤UNL)	3.34 (1.87–5.97)	<0.001 *	2.44 (1.2–4.96)	0.013 *
Baseline albumin (≥3.5 g/dL vs. <3.5 g/dL)	0.61 (0.36–1.05)	0.076		
Baseline sodium (≥135 mmol/L vs. <135 mmol/L)	0.38 (0.20–0.70)	0.002 *		
Baseline NLR (≥3.37 vs. <3.37)	2.94 (1.75–4.94)	<0.001 *		
Baseline PLR (≥146.4 vs. <146.4)	2.75 (1.60–4.72)	<0.001 *	1.79 (0.84–3.85)	0.133
Recent use of antibiotics within the past 30 days (yes vs. no)	1.72 (1.04–2.85)	0.036 *		
Differentiation (well or moderately vs. poorly differentiated)	1.37 (0.69–2.71)	0.374		
PD-L1 status				
CPS 22C3 (≥10 to <50 vs. ≥50)	3.44 (1.56–7.58)	0.002 *	2.46 (1.01–6.00)	0.047 *
CPS 22C3 (<10 vs. ≥50)	3.79 (1.6–8.96)	0.002 *	4.14 (1.51–11.32)	0.006 *

* Asterisks indicate statistically significant *p* values (< 0.05). HR, hazard ratio; CI, confidence interval; ICI, immune checkpoint inhibitor; ECOG PS, Eastern Cooperative Oncology Group performance status; Mb, megabase; Hb, hemoglobin; ALC, absolute lymphocyte count; ANC, absolute neutrophil count; CRP, C-reactive protein; UNL, upper normal limit; NLR, neutrophil–to–lymphocyte ratio; PLR, platelet–to–lymphocyte ratio; PD-L1, programmed death-ligand 1; CPS, combined positive score.

**Table 4 cancers-17-03149-t004:** Univariate and multivariate analysis for overall survival.

Variable	Univariable		Multivariable	
	HR (95% CI)	*p* Value	HR (95% CI)	*p* Value
Age (≥65 years vs. <65 years)	0.63 (0.33–1.20)	0.160		
ICI (nivolumab vs. pembrolizumab)	1.23 (0.66–2.28)	0.510		
Sex (female vs. male)	1.04 (0.46–2.34)	0.925		
No. of metastatic organs (≥2 vs. ≤1)	1.98 (1.09–3.59)	0.024 *		
Liver metastasis (yes vs. no)	3.26 (1.76–6.06)	<0.001 *		
Lung metastasis (yes vs. no)	1.36 (0.73–2.51)	0.332		
Peritoneum metastasis (yes vs. no)	2.65 (0.94–7.46)	0.064		
Bone metastasis (yes vs. no)	2.64 (1.32–5.25)	0.006 *		
Lymph node metastasis (yes vs. no)	0.85 (0.47–1.54)	0.587		
Cigarette use (never vs. former/current smoker)	0.97 (0.48–1.96)	0.927		
ECOG PS (≥2 vs. 0–1)	4.31 (2.11–8.78)	<0.001 *	2.90 (1.23–6.85)	0.015 *
Tumer mutation burden (≥20 Mb vs. <20 Mb)	1.06 (0.43–2.63)	0.897		
Baseline Hb (≥12 g/dL vs. <12 g/dL)	0.6 (0.33–1.09)	0.091		
Baseline ALC (≥1000/μL vs. <1000/μL)	0.71 (0.30–1.69)	0.441		
Baseline ANC (≥4000/μL vs. <4000/μL)	2.43 (1.16–5.06)	0.018 *		
Baseline CRP (>UNL vs. ≤UNL)	3.15 (1.62–6.13)	0.001 *	2.22 (1.06–4.66)	0.034 *
Baseline albumin (≥3.5 g/dL vs. <3.5 g/dL)	0.45 (0.24–0.82)	0.010 *		
Baseline sodium (≥135 mmol/L vs. <135 mmol/L)	0.37 (0.19–0.72)	0.004 *		
Baseline NLR (≥3.37 vs. <3.37)	2.52 (1.38–4.59)	0.003 *		
Baseline PLR (≥146.4 vs. <146.4)	2.58 (1.35–4.90)	0.004 *	2.15 (0.93–4.97)	0.072
Recent use of antibiotics within the past 30 days (yes vs. no)	2.20 (1.21–4.00)	0.009 *		
Differentiation (well or moderately vs. poorly differentiated)	0.99 (0.47–2.1)	0.987		
PD-L1 status				
CPS 22C3 (≥10 to <50 vs. ≥50)	3.47 (1.3–9.24)	0.013 *	2.28 (0.74–7.00)	0.152
CPS 22C3 (<10 vs. ≥50)	5.35 (1.91–15)	0.001 *	5.82 (1.78–19.03)	0.004 *

* Asterisks indicate statistically significant *p* values (<0.05). HR, hazard ratio; CI, confidence interval; ICI, immune checkpoint inhibitor; ECOG PS, Eastern Cooperative Oncology Group performance status; Mb, megabase; Hb, hemoglobin; ALC, absolute lymphocyte count; ANC, absolute neutrophil count; CRP, C-reactive protein; UNL, upper normal limit; NLR, neutrophil–to–lymphocyte ratio; PLR, platelet–to–lymphocyte ratio; PD-L1, programmed death-ligand 1; CPS, combined positive score.

## Data Availability

The data that support the findings of this study are available from the corresponding author upon reasonable request.

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
