# Peer review of "Real-World Outcomes and Biomarker Analysis Based on Routine Clinical, Laboratory, and Pathologic Parameters in Metastatic or Unresectable Esophageal Cancer Treated with First-Line Anti-PD-1 Plus Fluoropyrimidine and Platinum"

_cancers, 2025, doi:10.3390/cancers17193149_

Round 1
Reviewer 1 Report
Comments and Suggestions for Authors
- This is well written manuscript with interesting data. I have a few comments.
- Define all abbreviations as they appear in the text for the first time (such as ECOG, etc. ).
- This was mentioned in the discussion but I am wondering why MSI status was not evaluated in more than 50% patient samples? This is important characteristics associated with ICI response and it is easy to asses by IHC or PCR. This is clearly needed.
- Why is not baseline absolute neutrophil count included as one of the factors for PFS? Table 3 indicates it is significant factor.
- Spelling- tumor mutational burden (Table 3).
Author Response
Comments 1: This is well written manuscript with interesting data. I have a few comments.
Response 1: We greatly appreciate the reviewer’s positive feedback and constructive suggestions.
Comments 2: Define all abbreviations as they appear in the text for the first time (such as ECOG, etc. ).
Response 2: We have carefully reviewed the entire manuscript and ensured that all abbreviations, including ECOG and others, are defined at their first appearance in the text.
Comments 3: This was mentioned in the discussion but I am wondering why MSI status was not evaluated in more than 50% patient samples? This is important characteristics associated with ICI response and it is easy to asses by IHC or PCR. This is clearly needed.
Response 3: We thank the reviewer for highlighting the potential role of MSI status. We fully agree that MSI-H/dMMR tumors are associated with responsiveness to immune checkpoint inhibitors in several malignancies. However, in advanced or metastatic esophageal squamous cell carcinoma, MSI-high or dMMR tumors are known to be extremely rare, and in routine clinical practice, PD-L1 expression (CPS or TPS) is generally regarded as the key biomarker that guides treatment decisions rather than MSI status. Consequently, MSI testing has not been performed routinely in real-world practice. Since our study was conducted retrospectively, these practice patterns were reflected in our cohort, resulting in the absence of MSI data in more than half of the patients.
Comments 4: Why is not baseline absolute neutrophil count included as one of the factors for PFS? Table 3 indicates it is significant factor.
Response 4: We thank the reviewer for this important comment. In the univariate analysis, baseline absolute neutrophil count (ANC) was indeed associated with PFS. However, in constructing the multivariable analysis using the Cox proportional hazards model, we carefully considered the issue of multicollinearity. Because ANC, lymphocyte count, and platelet count are mathematically and biologically interrelated, including them simultaneously in the Cox model could result in multicollinearity and reduced model stability. Therefore, instead of including these individual parameters separately, we used the neutrophil-to-lymphocyte ratio (NLR) and the platelet-to-lymphocyte ratio (PLR), which integrate these variables and are widely recognized as robust indicators of systemic inflammation and prognosis.
We have clarified this rationale in the revised Methods section (page 5, lines 190–195).
“For hematologic parameters, absolute neutrophil count, lymphocyte count, and platelet count were not included separately, as including these variables simultaneously in the Cox model could lead to multicollinearity. Instead, the neutrophil-to-lymphocyte ratio (NLR) and platelet-to-lymphocyte ratio (PLR) were used, as these composite indices inte-grate the relevant measures and are widely recognized as comprehensive indicators of systemic inflammation and prognosis.”
In addition, the Results section has been revised to explicitly note that baseline ANC was significant in the univariate analysis but was not entered into the multivariable model for the reasons described above (page 10, line 297-298).
Comments 5: Spelling- tumor mutational burden (Table 3).
Response 5: We appreciate the reviewer’s careful observation. The spelling of “tumor mutational burden” in Table 3 has been corrected to ensure consistency throughout the manuscript.
Reviewer 2 Report
Comments and Suggestions for Authors
The authors retrospectively analyzed the outcomes of 87 patients with metastatic or unresectable ESCC who received first-line treatment with anti-PD-1 plus chemotherapy. They identified ECOG performance status ≥2, elevated CRP, and lower PD-L1 CPS as poor prognostic factors and proposed a simple scoring system based on these variables to stratify patient prognosis. However, the use of anti-PD-1 plus chemotherapy as first-line treatment for metastatic or unresectable ESCC is already recommended by clinical guidelines, and thus its efficacy does not require validation through a retrospective analysis of 87 patients. Moreover, all three prognostic factors presented by the authors are rather obvious. It is self-evident that poor ECOG performance status leads to worse outcomes; elevated CRP is already well recognized as a poor prognostic factor in most cancers or simply a reflection of advanced disease; and PD-L1 CPS itself is an established biomarker that guides treatment decisions.
Author Response
Comments 1: The authors retrospectively analyzed the outcomes of 87 patients with metastatic or unresectable ESCC who received first-line treatment with anti-PD-1 plus chemotherapy. They identified ECOG performance status ≥2, elevated CRP, and lower PD-L1 CPS as poor prognostic factors and proposed a simple scoring system based on these variables to stratify patient prognosis. However, the use of anti-PD-1 plus chemotherapy as first-line treatment for metastatic or unresectable ESCC is already recommended by clinical guidelines, and thus its efficacy does not require validation through a retrospective analysis of 87 patients. Moreover, all three prognostic factors presented by the authors are rather obvious. It is self-evident that poor ECOG performance status leads to worse outcomes; elevated CRP is already well recognized as a poor prognostic factor in most cancers or simply a reflection of advanced disease; and PD-L1 CPS itself is an established biomarker that guides treatment decisions.
Response 1: We sincerely thank the reviewer for this thoughtful comment regarding the novelty of our work. We agree that ECOG performance status, CRP, and PD-L1 CPS have individually been reported as prognostic factors in cancer, including esophageal cancer. Indeed, it is self-evident that poor ECOG performance status and systemic inflammation reflected by elevated CRP are associated with inferior outcomes, and PD-L1 CPS is an established biomarker guiding immunotherapy.
However, our study has several points of clinical and academic relevance:
- Real-world evidence: While pivotal clinical trials such as KEYNOTE-590 and CheckMate-648 have established anti-PD-1 plus chemotherapy as the standard first-line regimen for advanced ESCC, real-world data reflecting routine practice remain limited. Our multi-institutional cohort provides such evidence by focusing on real-world patients, many of whom had poor ECOG performance status, thereby reflecting routine clinical practice.
- Integrated prognostic scoring system: Although each factor is known, their combination provides a more comprehensive and robust assessment of patient prognosis. We hypothesized that integrating these distinct aspects—host performance (ECOG), systemic inflammatory response (CRP), and tumor biology (PD-L1 CPS)—would offer superior prognostic stratification compared to any single factor alone. This pragmatic model may allow physicians to identify patients at higher risk of poor outcomes using readily available parameters.
- Clinical utility: Beyond reiterating known factors, our score offers a tool that may guide treatment discussions, inform risk-adapted monitoring, and allow physicians to identify patients who may achieve long-term survival with the standard first-line regimen, thus potentially avoiding unnecessary toxicities or alternative treatments.
In summary, although the individual prognostic variables are not novel, the integration of these factors into a real-world, clinically applicable scoring system represents the unique contribution of our study. We believe this approach enhances its practical value and complements existing trial data by reflecting outcomes in daily clinical practice.
Reviewer 3 Report
Comments and Suggestions for Authors
Thank you for sending your manuscript, this surely entailed a considerably amount of work. It denotes a thoroughly ran study, well thought and well implemented. The manuscript is well written and brings a new prognostic scoring model. However, I have found some points I would like you to comment on, please:
Line 128 – “ICI” – please write in full any abbreviation the first time you use it in the manuscript (even if it is included in the abbreviations section).
Line 208 – Figure 1 – I would like you to comment on the different results you obtained for CPS (22C3) vs CPS (28-8) and TPS (23C3) vs TPS (28-8). I understand that the number should be different, but the percentage is very different between the two kits used. Why do you think that has happened?
Line 208 – Figure 1 – “Prior treatment for esophageal cancer” you state “25 (28.7%)”. Did only 25 patients have prior treatment? If so, how do you explain Surgery – 21, Concurrent chemoradiotherapy – 15, Radiation therapy – 2, Chemotherapy – 13? 21+15+2+13 is not 25.
Line 223 – “CPS (22C3) results were available in 83 patients (95.4%), of whom 77.1% had a CPS ≥10.” – accordingly to figure 1, it is 73.6%. When you mention “of whom” it is supposed the percentage being related to the number of patients having the test, right?
Line 224 – “CPS (28-8) results were available in 34 patients (39.1%), of whom 55.9% showed a CPS ≥10” – accordingly to figure 1, it is 21.8%
Line 225 – “TPS (22C3) results were available in 86 patients (98.9%), of whom 79.1% had a TPS ≥1%”– accordingly to figure 1, it is 78.2%
Line 226 – “TPS (28-8) results were available in 50 patients (57.5%), of whom 88.0% showed a TPS ≥1%” – accordingly to figure 1, it is 50.6%
Line 286 – You do not mention in this paragraph that baseline ANC were also significantly associated with shorter PFS (which is not accordingly with your table 3, in which it shows a p value of 0.002 for univariate analysis).
Author Response
Comments 1: Line 128 – “ICI” – please write in full any abbreviation the first time you use it in the manuscript (even if it is included in the abbreviations section).
Response 1: We thank the reviewer for this comment. We would like to clarify that the abbreviation “ICI” (immune checkpoint inhibitor) is first introduced in full at line 62-63 of the manuscript, and we have ensured that it is written in full at its first occurrence.
Comments 2: Line 208 – Figure 1 – I would like you to comment on the different results you obtained for CPS (22C3) vs CPS (28-8) and TPS (23C3) vs TPS (28-8). I understand that the number should be different, but the percentage is very different between the two kits used. Why do you think that has happened?
Response 2: We thank the reviewer for this important comment. At first glance, the percentages of PD-L1 expression appear discrepant between the 22C3 and 28-8 assays. For example, in the overall cohort of 87 patients, 73.6% were classified as CPS ≥10 by 22C3, whereas only 21.8% were classified as CPS ≥10 by 28-8. However, this apparent discrepancy largely reflects the fact that PD-L1 testing was not performed uniformly in all patients.
In detail, CPS by 22C3 was available in 83 patients, of whom 64 (77.1%) had CPS ≥10. CPS by 28-8 was available in only 34 patients, of whom 19 (55.9%) had CPS ≥10. Similarly, TPS by 22C3 was available in 86 patients, of whom 68 (79.1%) had TPS ≥1%. TPS by 28-8 was available in 50 patients, of whom 44 (88.0%) had TPS ≥1%. Thus, the differences in overall percentages mainly reflect the different denominators due to assay availability, rather than true biological discordance.
To further address this point, we analyzed the subset of patients who had both assays available. For CPS, we found a strong correlation between 22C3 and 28-8 (Pearson r ≈ 0.79, Spearman ρ ≈ 0.83). For TPS, the two assays showed an even higher correlation (Pearson r ≈ 0.92, Spearman ρ ≈ 0.81). Scatter plots with clinically relevant cutoffs (CPS=10 and TPS=1%) demonstrated that most patients were classified consistently across assays, with only a minority of discrepant cases.
Taken together, the apparent differences in percentages are mainly attributable to patient selection and missing data inherent to the retrospective, real-world nature of our study. When the same patients were tested with both assays, 22C3 and 28-8 yielded broadly concordant results for both CPS and TPS.
Comments 3: Line 208 – Figure 1 – “Prior treatment for esophageal cancer” you state “25 (28.7%)”. Did only 25 patients have prior treatment? If so, how do you explain Surgery – 21, Concurrent chemoradiotherapy – 15, Radiation therapy – 2, Chemotherapy – 13? 21+15+2+13 is not 25.
Response 3:
We thank the reviewer for carefully pointing out this issue. The discrepancy arises because some patients received more than one type of prior treatment before starting first-line palliative chemotherapy. For example, a patient may have undergone surgery and, upon recurrence, subsequently received induction chemotherapy followed by definitive concurrent chemoradiotherapy (CCRT), thereby undergoing three different modalities. Similarly, some patients underwent two types of treatment, such as preoperative chemotherapy followed by surgery.
In our cohort, 8 patients received two different prior treatments and 9 patients received three different prior treatments. Therefore, although the numbers in table 1 are shown separately by treatment modality (Surgery: 21; CCRT: 15; Radiation therapy: 2; Chemotherapy: 13), these categories overlap. When accounting for patients who underwent multiple treatments (21 + 15 + 2 + 13 − 8 − 2×9), the total number of patients with any prior treatment is 25 (28.7%), which matches the table.
We have also added the following note as a footnote to Table 1 (page 7, lines 221-222) :
“The numbers of patients by treatment modality are not mutually exclusive because some patients received more than one prior treatment.”
Comments 4: Line 223 – “CPS (22C3) results were available in 83 patients (95.4%), of whom 77.1% had a CPS ≥10.” – accordingly to figure 1, it is 73.6%. When you mention “of whom” it is supposed the percentage being related to the number of patients having the test, right?
Response 4: As correctly noted by the reviewer, the percentage in the text refers to the number of patients who actually had CPS (22C3) results available. In Table 1, the percentage was calculated based on the entire cohort of 87 patients, where 64 patients had CPS ≥10 (64/87 = 73.6%). In contrast, the sentence in the Results section refers specifically to the 83 patients with available CPS (22C3) results, among whom 64 patients had CPS ≥10 (64/83 = 77.1%).
Comments 5: Line 224 – “CPS (28-8) results were available in 34 patients (39.1%), of whom 55.9% showed a CPS ≥10” – accordingly to figure 1, it is 21.8%
Response 5: We agree with the reviewer that the percentages differ depending on the denominator used. In Table 1, the percentage was calculated based on the entire cohort of 87 patients, where 19 patients had CPS (28-8) ≥10 (19/87 = 21.8%). In contrast, the sentence in the Results section refers specifically to the 34 patients with available CPS (28-8) results, among whom 19 patients had CPS ≥10 (19/34 = 55.9%).
Comments 6: Line 225 – “TPS (22C3) results were available in 86 patients (98.9%), of whom 79.1% had a TPS ≥1%”– accordingly to figure 1, it is 78.2%
Response 6: The apparent difference arises from the denominator used for the calculation. In Table 1, the percentage was based on the entire cohort of 87 patients, where 68 patients had TPS (22C3) ≥1% (68/87 = 78.2%). In contrast, the statement in the Results section refers specifically to the 86 patients with available TPS (22C3) results, among whom 68 patients had TPS ≥1% (68/86 = 79.1%).
Comments 7: Line 226 – “TPS (28-8) results were available in 50 patients (57.5%), of whom 88.0% showed a TPS ≥1%” – accordingly to figure 1, it is 50.6%
Response 7: This difference is due to the denominator used for the calculation. In Table 1, the percentage was calculated based on the entire cohort of 87 patients, where 44 patients had TPS (28-8) ≥1% (44/87 = 50.6%). In contrast, the statement in the Results section refers specifically to the 50 patients with available TPS (28-8) results, among whom 44 patients had TPS ≥1% (44/50 = 88.0%).
Comments 8: Line 286 – You do not mention in this paragraph that baseline ANC were also significantly associated with shorter PFS (which is not accordingly with your table 3, in which it shows a p value of 0.002 for univariate analysis)
Response 8: We thank the reviewer for this important comment. In the univariate analysis, baseline absolute neutrophil count (ANC) was indeed associated with PFS. However, in constructing the multivariable analysis using the Cox proportional hazards model, we carefully considered the issue of multicollinearity. Because ANC, lymphocyte count, and platelet count are mathematically and biologically interrelated, including them simultaneously in the Cox model could result in multicollinearity and reduced model stability. Therefore, instead of including these individual parameters separately, we used the neutrophil-to-lymphocyte ratio (NLR) and the platelet-to-lymphocyte ratio (PLR), which integrate these variables and are widely recognized as robust indicators of systemic inflammation and prognosis.
We have clarified this rationale in the revised Methods section (page 5, lines 190–195).
“For hematologic parameters, absolute neutrophil count, lymphocyte count, and platelet count were not included separately, as including these variables simultaneously in the Cox model could lead to multicollinearity. Instead, the neutrophil-to-lymphocyte ratio (NLR) and platelet-to-lymphocyte ratio (PLR) were used, as these composite indices inte-grate the relevant measures and are widely recognized as comprehensive indicators of systemic inflammation and prognosis.”
In addition, the Results section has been revised to explicitly note that baseline ANC was significant in the univariate analysis but was not entered into the multivariable model for the reasons described above (page 10, lines 297-298).
Round 2
Reviewer 2 Report
Comments and Suggestions for Authors
As this manuscript was previously rejected for fundamental reasons that cannot be addressed, I do not consider further review appropriate.